# FAST CONDITIONAL INTERVENTION IN ALGORITHMIC RECOURSE WITH REINFORCEMENT LEARNING

## ABSTRACT

Explaining the decisions made by machine learning classifiers aids individuals in identifying critical factors and charting future plans. Recent studies have shown that incorporating causal graphs of input features provides more realistic explanations; however, this also introduces new challenges such as handling noisy graphs and efficiently performing inference with black-box classifiers. In this work, we tackle these issues by presenting an efficient reinforcement learning (RL)-based approach with an idea of conditional intervention. Our intervention method is theoretically preferable and considers both feature dependencies and incompleteness of graphs. Simultaneously, the RL-based method offers the capacity to learn the intervention process while guarantees computational complexity at inference stage. In the experiments, we showcase the efficiency and superior performance of our solution when compared to baseline methods on both synthetic and real datasets.

## 1 INTRODUCTION

Explaining machine learning models in decision making can be beneficial in practice. For example, a customer who is rejected by a loan approval system may expect not only the decision but also some advice for getting approval next time. Relevant topics are counterfactual explanation and algorithmic recourse. Counterfactual explanation or CE (Wachter et al., 2017) finds a minimum feature set that can be intervened to change the classifier output. Algorithmic recourse or AR (Mahajan et al., 2019; Verma et al., 2022; Kanamori et al., 2020), on the other hand, is based on CE but puts more attention on avoiding impractical suggestions (e.g., reducing age). A recent trend for achieving practical explanations is considering constraints or dependency between features. For example, Mahajan et al. (2019) propose to introduce structural causal models (SCMs) into the CE algorithm. OrdCE (Kanamori et al., 2021) finds explanations by solving an optimization problem constrained by causal graphs. As an accurate causal graph can be difficult to obtain, Karimi et al. (2020) propose Bayesian-based approaches to train models with incomplete structural equations. However, the approach requires the decision maker to be differentiable, or brute-force search and feature discretization are needed, making it infeasible and suboptimal in general.

In this work, we propose **C**onditional **I**ntervention for **A**lgorithmic **R**ecourse (CIAR) for efficiently generating explanations with incomplete causal graphs. The idea of conditional intervention is motivated by the existence of unobserved factors. An example is illustrated in Figure 1, where the unobserved factors, muscle and bone densities, exist and can offer extra paths for influencing weight.

This setting brings two challenges. First, the existence of the unobserved factors is unkown. Second, the extra intervention needs to depend on existing causal effects so that the desired output can be achieved (i.e., +2 kg on weight considering height to achieve +8 kg). To do this, we propose to formulate the intervention on weight by $P(do(\text{weight}) \mid \text{height})$ and attempt to approximate the distribution by learning from data. Intuitively, the estimated distribution implies the impact of possibly unobserved factors and a reasonable range for intervention. Compared with Karimi et al. (2020), our problem setting is more realistic yet challenging as the uncertainty of intervention are considered.

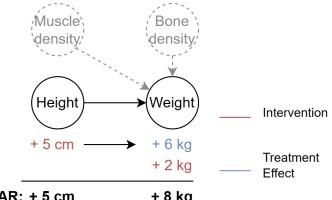

Figure 1: A causal graph of height and weight with unobserved factors, muscle and bone density.

To generate explanations with conditional interventions, we propose a novel reinforcement learning (RL) approach featuring a stable and efficient policy network. The most relevant work to ours could be FastAR (Verma et al., 2022). FastAR is a RL method focusing on fast AR generation given a SCM; however, we note that the intervention on endogenous features is not well handled. Therefore, the causality can be overwritten. Also, we find FastAR can suffer from the discretized action space and the encoding of categorical features.

The advantages of our model is summarized as follows.

- The proposed idea, conditional intervention, shares the same motivation with (Mahajan et al., 2019) but improves the formulation. Our objective function derived from the proposed intervention is theoretically more general and optimal.
- The experiments on both synthetic and real datasets show that CIAR outperforms other competitors and can significantly improve the efficiency. Specifically, CIAR is around 15 times faster than FastAR when generating explanations.
- CIAR has low requirements to use. The proposed algorithm does not assume complete causal graphs and can work with mixtures of numerical and categorical features. Also, CIAR can explain black-box classifiers without sacrificing efficiency.

## 2 PRELIMINARIES

### 2.1 PROBLEM STATEMENT

We first introduce the concept of parametrized action space, which is crucial to the implementation of interventions in AR with RL. Then we describe the setting of the causal models that we use to ensure causality between features after interventions. Finally we define our proposed concept of conditional intervention and clearly formulate the RL-based process of interventions for AR.

**Parametrized action space of AR with RL**  Solving AR problems with RL is non-trivial. At each step, the RL agent determines $W$ (*which feature* to intervene upon) and $V$ (*what value* to intervene to). In the researches of RL, Masson et al. (2016) formalized this kind of action as *parametrized actions* with the action space $A = \bigcup_{w \in A_d} \{(w, v) \mid v \in \Omega_w\}$, where every action in $A$ contains a discrete action $w \in A_d$ and a continuous action $v \in \Omega_w$. For AR tasks, $A_d$ is the set of all actionable features. $\Omega_w$ is the domain of feature $X_w$.

**Causality and intervention in AR**  To ensure interventions in AR respect the causal relationships between features, it requires a structural causal model (SCM) (Pearl, 2009).

**Definition 1** (Structure Causal Model). A structural causal model (SCM) is an ordered triple $< G, H, F >$, where $G$ is the set of exogenous features (variables); $H$ is the set of endogenous features; and $F$ is the set of structural equations that determines the values of $H$ from other features in $G \cup H$. For each $X_i \in H$ and $F_i \in F$, $X_i = F_i(PA_i, \mathcal{E}_i)$, where $PA_i$ denotes the parents of $X_i$ and $\mathcal{E}_i$ is the randomness that depicts the stochastic mapping from $PA_i$ to $X_i$.

A causal model can be represented by a directed acyclic graph (DAG). Figure 2 is an example of a causal model in DAG form. In Figure 2, $\{U_0, X_3, X_4\} \subset H$ with others in $G$. In an SCM, when a feature is intervened upon, it will have treatment effects on its descendants (endogenous). However, this does not imply the endogenous features themselves cannot be intervened upon. In the real world, it is unlikely we collect all the causes (parents) of an endogenous feature. We can have **unobserved causes** and that is where $\mathcal{E}_i$ in Definition 1 arises from. Figure 2 demonstrates this concept. We could lose $U_0, U_1, U_2$ in the causal model, so $\{X_3, X_4\} \subset H$ with others in $G$. When $X_3$ is intervened upon with its parent $X_2$ fixed, it actually means that the unobserved causes $U_0, U_2$ are intervened upon implicitly. Therefore, we can still have an intervention on an endogenous feature. We should determine this intervention according to the extent it is determined by its parents. The less it is determined by their parents, the more *space* we can intervene.

**Definition 2** (Conditional intervention). In comparison with regular intervention $do(X_i = x_i')$, where $x_i'$ is sampled from the sample space $\Omega(X_i)$, conditional intervention $do(X_i = x_i' \mid pa_i)$ has the intervention value $x_i'$ sampled from the distribution $p(X_i \mid pa_i)$. Then we assume

$$Pr(X_i = x_i' \mid pa_i) \geq \text{threshold}, \tag{1}$$

which means given the parent values of $X_i$, the value of of intervention, $x_i'$, should be a value that is likely to be observed (i.e, above a pre-determined threshold).

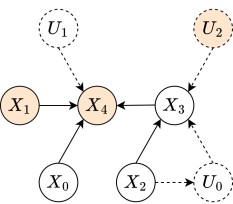

Figure 2: An example of causal graph. Categorical features are labeled with orange color. Features with dashed line are assumed unobserved.

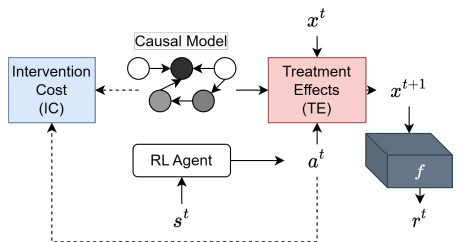

Figure 3: The illustration of CIAR.

**Problem formulation** Given an instance (input features) $\boldsymbol{x}$, a classifier $f$, the target class $y'$ and the SCM that describes the causal relationships between features, our goal is to generate, with RL, **a sequence of conditional interventions** $\{a^t = (w^t, v^t); t = 0, \ldots, T-1\}$ that in conjunction with the SCM, maps $\boldsymbol{x}$ to its counterfactual instance (AR) $\boldsymbol{x}'$ s.t. the following requirements are satisfied.

1. $f(\boldsymbol{x}') = y'$.
2. $T$ (number of steps) $\leq p$ (number of actionable features).
3. $\boldsymbol{x}'$ preserves causal relationships between features.
4. $\sum_{t=0}^{T-1} \|v^t - x_w^t\|$ is minimized, where $x_w^t$ is the feature value $v^t$ is to replace.

## 2.2 RELATED WORK

Most previous works formulate AR (CE) problems as solving optimization problems at inference stage (Wachter et al., 2017; Mothilal et al., 2020; Karimi et al., 2019; Kanamori et al., 2020). For better efficiency, Nemirovsky et al. (2020) and Pawelczyk et al. (2020) propose generative methods. Methods above provide the final AR results without clear reasoning. RL-based AR methods (Verma et al., 2022; Chen et al., 2022), on the other hand, are attractive in providing a clear sequence of actions for users to follow and are more efficient than the optimization-based methods (Wachter et al., 2017; Mothilal et al., 2020; Karimi et al., 2019; Kanamori et al., 2020). In the context of RL researches, to deal with the parametrized action space for AR, with most RL algorithms, one should unify the action spaces of the two sets of actions (Sherstov & Stone, 2005), leading to compromise in performance. Wei et al. (2018) proposed a hierarchical approach PASVG(0) for parametrized action space, where a discrete action is firstly determined, and a continuous action is then determined by conditioning on the known discrete action. The other work, P-DQN (Xiong et al., 2018), first determines the continuous action explicitly. Afterward, the Q network examines all the discrete actions to find the one that is the most consistent with the continuous action. The latter is not as efficient due to the nature of DQN (Mnih et al., 2013), having to examine all the discrete actions. For RL-based AR methods, FastAR (Verma et al., 2022) discretized the action space, which could hamper the performance. ReLAX (Chen et al., 2022) solved the parametrized action space using P-DQN (Xiong et al., 2018). Neither FastAR nor ReLAX directly decode categorical features.

To preserve causality in AR tasks, Mahajan et al. (2019) proposed incorporating the SCM and minimizing a causal proximity for endogenous features, which can be applied in limited cases where the conditional (on parents) features are close to the likes of normal distributions. However, even with normality assumption, the minimizer to this proximity is still not optimal for causality (as we derived the optimum loss in section 3.1). Kanamori et al. (2021) proposed OrdCE that optimizes the *active actions* by generating ordered interventions given the SCM, solved by mixed integer programming. It only works for linear SCMs and non-blackbox classifiers. Karimi et al. (2020) enforces the probabilistic treatment effects of interventions by learning the interventional distributions with CVAE (Sohn et al., 2015). Verma et al. (2022) proposed FastAR, an RL-based algorithm that also enforces the treatment effects. For Karimi et al. (2020) and Verma et al. (2022), intervention on endogenous features is not conditional on parents, which could overwrite the causality. On the other hand, disallowing intervention on endogenous features greatly limits the search space of AR.

## 3 METHODOLOGY

**Overview**  Figure 3 illustrates our algorithm. At step $t$, the RL agent determines the intervention $a^t$ based on state $s^t$. The intervention cost (IC) helps ensure each intervention on an endogenous feature is a conditional intervention as in definition 2. The RL transition function enforces the treatment effects (TE) on the descendants of the intervened feature to preserve the causal relationships. The intervened feature values $\boldsymbol{x}^{t+1}$ will query the classifier $f$ and get a reward. This process repeats until the classifier output is changed to the target class or all features are intervened upon. An example is in figure 5, which we will explain in the end of the section. In Section 3.1, we introduces IC for AR methods. Section 3.2 elaborates the RL agent with the focus on the parametrized action space, the RL transition function combined with TE, and the incorporation of IC to the RL agent.

### 3.1 INTERVENTION COST (IC)

Intervention cost is a novel, differentiable loss function to help find the intervention value $x_i'$ for endogenous feature $X_i$ that satisfies definition 2. It finds out to what extent $X_i$ is determined by its parents. We intervene upon $X_i$ only if it is *not* fully determined by its parents. We can intervene upon multiple features. To ensure a *likely* intervention profile, we consider the joint distribution of all features $p(x_1, \ldots, x_p)$. With *Causal Markov condition* (Hausman & Woodward, 1999),

$$p(x_1, \ldots, x_p) = p(\boldsymbol{x}_{exo})\Pi_{X_i \in endo}p(x_i \mid pa_i), \tag{2}$$

where $exo$ and $endo$ denotes exogenous and endogenous features respectively.
Exogenous features are not constrained by parents. Let $\boldsymbol{x}'$ be the intervened feature vector. We consider the following likelihood function.

$$p(\boldsymbol{x}'_{endo} \mid \boldsymbol{x}'_{exo}) = \Pi_{X_i \in endo}p(x_i' \mid pa_i'). \tag{3}$$

The solution to $\boldsymbol{x}'_{endo}$ is supposed to maximize this likelihood function given $\boldsymbol{x}'_{exo}$. Hence, we minimize the negative log-likelihood

$$-\log p(\boldsymbol{x}'_{endo} \mid \boldsymbol{x}'_{exo}) = - \sum_{X_i \in endo} \log p(x_i' \mid pa_i'). \tag{4}$$

**Numerical case**  Let's first consider the case where

$$X_i \mid pa_i' \sim \mathcal{N}(\mu(X_i \mid pa_i'), \sigma(X_i \mid pa_i')), \tag{5}$$

i.e., $X_i \mid pa_i'$ follows Gaussian distribution where its mean and standard deviation depends on its parent values. In this case, from Eq. 4, the intervention cost for intervening upon $X_i$ to $x_i'$ is

$$IC(do(X_i = x_i' \mid pa_i'), pa_i') = (\frac{x_i' - \mu(X_i \mid pa_i')}{\sqrt{2}\sigma(X_i \mid pa_i')})^2 + \log \sigma(X_i \mid pa_i'). \tag{6}$$

From Eq. 6, we also see that the causal proximity from Mahajan et al. (2019), which only considers the first conditional moment, is not optimum even under normality assumption. For non-Gaussian cases, while we may estimate the conditional distribution $p(X_i \mid pa_i)$ using generative models or kernel density estimation (Rosenblatt, 1956), it is extremely demanding for both data quality and quantity, especially when $pa_i$ is high-dimensional (multiple parents features). Instead, motivated by Eq. 6, we make use of the conditional mean and conditional variance, reducing the problem from estimating infinite number of moments to estimating the first and second moments. We construct an interval

$$I_{x_i'|pa_i',\kappa} = \{x_i' : |x_i' - \mu(X_i \mid pa_i')| \leq \sigma(X_i \mid pa_i') \cdot \kappa\} \tag{7}$$

for $x_i'$ to reside in. By Chebyshev's inequality, $Pr(X_i \mid pa_i' \notin I_{x_i'|pa_i',\kappa}) \leq \frac{1}{\kappa^2}$. That is, given parent values, $I_{x_i'|pa_i',\kappa}$ captures a suitable range for $X_i$ despite its distribution. $\mu(X_i \mid pa_i')$ and $\sigma(X_i \mid pa_i')$ can be easily estimated without assumptions on distributions. We leave the details to the Appendix. Finally, in general cases, the intervention cost for the conditional intervention $do(X_i = x_i' \mid pa_i')$ is

$$IC_\kappa(do(X_i = x_i' \mid pa_i'), pa_i') = \max(0, |\frac{x_i' - \mu(X_i \mid pa_i')}{\sigma(X_i \mid pa_i')}| - \kappa)^2, \tag{8}$$

minimizing which will enforce $x_i'$ fall into the interval $I_{x_i'|pa_i',\kappa}$, so we can ensure $do(X_i = x_i' \mid pa_i')$ to have the property in Eq. 1.

**Categorical case**   When we need to intervene upon a categorical endogenous feature $X_i$, the AR methods model a discrete distribution $P_m(X_i')$ and sample $x_i'$ it. We want this distribution close to $P(X_i \mid pa_i')$. In discrete case, $P(X_i \mid pa_i')$ can be easily estimated with multi-class classifiers. The generalization of intervention cost to a categorical endogenous feature $X_i$ is

$$IC(do(X_i \sim P_m(X_i') \mid pa_i'), pa_i') = \mathrm{D_{KL}}(P(X_i \mid pa_i')\|P_m(X_i')), \qquad (9)$$

where $\mathrm{D_{KL}}$ denotes Kullback–Leibler divergence (Csiszar, 1975). The notation $do(X_i \sim P_m(X_i') \mid pa_i')$ denotes that given parent values, the intervention value on $X_i$ is sampled from $P_m(X_i')$.

**Remark 1.** The greater uncertainty of $X_i \mid pa_i'$ corresponds to the smaller intervention cost.

In numerical cases, we can measure the uncertainty of $X_i \mid pa_i'$ by $\sigma(X_i \mid pa_i')$. It can be seen from Eq. 8 that greater **conditional standard deviation** $\sigma(X_i \mid pa_i')$ leads to smaller intervention cost.

In categorical cases, Eq. 9 can also be expressed by

$$IC(do(X_i \sim P_m(X_i') \mid pa_i'), pa_i') = \mathrm{H}(P(X_i \mid pa_i')), P_m(X_i')) - \mathrm{H}(P(X_i \mid pa_i')), \qquad (10)$$

where $\mathrm{H}(\cdot)$ and $\mathrm{H}(\cdot, \cdot)$ denote entropy and cross entropy respectively. Hence, the intervention cost decreases as the uncertainty (**conditional entropy**) of $P(X_i \mid pa_i')$ increases.

The total intervention cost sums over all endogenous features, so **the optimization process in AR could find out which endogenous features are less determined by its parents (higher uncertainty) and focuses on intervening them if they are important to the classifier output.**

One thing to notice is that if one would like to optimize $IC(do(X_i \sim P_m(X_i') \mid pa_i'), pa_i')$ through its gradient, optimize $g(IC(do(X_i \sim P_m(X_i') \mid pa_i'), pa_i'))$ instead, where $g$ is a non-linear increasing function in $[0, \infty]$. This is to prevent vanishing of $\mathrm{H}(P(X_i \mid pa_i'))$ in the gradient.

### 3.2   The RL agent

At each step, the RL agent determines an intervention on the original instance $\boldsymbol{x}$. After several interventions, the modified input $\boldsymbol{x}'$ is expected to lead to the target classifier output. The process can be formulated as a Markov Decision Process (MDP).

#### 3.2.1   Notation

We denote a random feature vector by $X$ and its realization by $\boldsymbol{x}$. The subscript $X_i$ or $x_i$ denotes the $i^{th}$ dimension of the vector. $X_i$ also represents the $i^{th}$ feature in the dataset. The superscript $X^t$ or $\boldsymbol{x}^t$ means the vector at step $t$. $w_b^t$ is the action record, which records the features ever chosen by the agent before step $t$. $\mathbb{M}$ and $\mathbb{C}$ represent the set of the numerical and categorical features respectively. We assume the number of actionable feature $p$.

#### 3.2.2   The MDP specification

**Actions:**   The actions (interventions) are represented by a sequence $\{a^t = (w^t, v^t) \mid t = 0, 1, ..., T-1\}$, where $w^t$ is the feature intervened upon and $v^t$ is the value of intervention.

**States:**   Since we intervene upon each feature at most once, in order to control the actions, the policy should be aware of the action record $w_b^t = \{w^0, ..., w^{t-1}\}$ at step $t$. Hence, the state $s^t = (x^t, w_b^t)$.

**Transition function and treatment effects:**   The transition function $\Phi(s^t, a^t)$ maps state $s^t$ and action $a^t$ to the next state $s^{t+1}$. Assume $w^t = X_k$. Then $x_k^{t+1} = v^t$. The chosen feature $X_k$ is added to the action record, $w_b^{t+1} = w_b^t \cup \{X_k\}$. In the language of causality, we have $do(X_k = v^t)$ or $do(X_k = v^t \mid pa_k^t)$. The later occurs when $X_k$ is endogenous. Also, when feature $X_k$ is intervened upon, it will have *treatment effects* on its descendants; hence the descendants will also be updated by

$$x_i^{t+1} = x_i^t - f_i(pa_i^t) + f_i(pa_i^{t+1}) \forall X_i \in \mathrm{desc}(X_k), \qquad (11)$$

from Pearl's abduction-action-prediction (Pearl, 2009). Note that we should update $X_i \in \mathrm{desc}(X_k)$ when all the features in $\mathrm{parents}(X_i) \cap \mathrm{desc}(X_k)$ have updated. Hence, the order of update of $X_i$ when $X_k$ is intervened upon is calculated by the longest path length between $X_i$ and $X_k$. It's recommended to **compute pair-wise longest path lengths with Floyd-Warshall algorithm**.

**Reward function:** Without loss of generality, we assume a binary classifier $f(\boldsymbol{x}) \in \{0, 1\}$ and the target class is 1. At step $t$, the reward for action $(w^t, v^t)$ when observing $s^t$ is

$$r^t = f(x^{t+1}) - \lambda \sum_{i=0}^{t} \text{dist}(v^t, x_w^t), \tag{12}$$

where $x_w^t$ is the feature value that $v^t$ will replace; $\lambda$ controls the scale of interventions. We can also have pre-defined ranges for features, e.g. $education\ level \in [0, 20]$ or pre-defined constraints, e.g. $height$ cannot decrease. If $(w^t, v^t)$ fails the constraints, set $r^t = -5$. We optimize the cumulative reward $R^t = \Sigma_{i=t}^{T-1} \gamma^{i-t} r^t$, where $\gamma$ is the discount factor.

### 3.2.3 A STABLE AND EFFICIENT POLICY NETWORK

The policy network $\Pi$ models the action space discussed in Section 2.1 by the joint distribution of $A^t = (W^t, V^t)$. We follow the hierarchical sampling mechanism in PASVG(0) (Wei et al., 2018). Although PASVG(0) can be unstable due to the joint-learning between the discrete action policy and parameter policy, pointed out by the authors, we found some architectural corrections will fix it. The network guarantees number of accesses to the classifier as $O(p)$, which to our knowledge, no other iterative methods does.

**Output:** Given state $s^t$, to determine $a^t = (w^t, v^t)$, $\Pi$ outputs the joint distribution of random vector $A^t = (W^t, V^t)$. The distribution of $A^t$ can be expressed as

$$P(A^t) = P(W^t)P(V^t \mid W^t). \tag{13}$$

We first sample $w^t$ (which feature) from $P(W^t)$, and then sample $v^t$ from $P(V^t \mid W^t = w^t)$ to determine the value. Figure 4 is the model architecture of the agent. The sub-network $\text{MLP}_w$ outputs the parameters of a multinomial distribution $\psi_w^t$ to model $\text{P}(W^t)$. The dimension of $\psi_w^t$ is $p$. The $i^{th}$ element of $\psi_w^t$ represents the probability that feature $X_i$ is sampled. We allowed each feature to be chosen at most once. For features in the action record $w_b^t$, we set their probability to 0. We achieve this by masked softmax.

$$m_i^t = \begin{cases} 1, & \text{if } X_i \in w_b^t, \\ 0, & \text{otherwise} \end{cases} \qquad \psi_w^t = \frac{\text{SoftMax}(l^t) \odot m^t}{\text{SoftMax}(l^t) \cdot m^t}, \tag{14}$$

where $l^t$ is the logits output by the network and $m_i^t$ is the $i^{th}$ element of the mask $m^t$; $\odot$ denotes element-wise product. Different from ReLAX (Chen et al., 2022), our RL agent is based on a policy network, which does not require examining the rewards for all discrete actions. Therefore, CIAR guarantees the number of accesses to classifier as $O(p)$.

After deciding $w^t$, $\text{MLP}_v$ takes $\psi_w^t$ and $w^t$ as inputs and outputs the distribution parameters $\psi_v^t$ to model $P(V^t \mid W^t = w^t)$. Assume the sampled feature is $X_k$ ($w^t = X_k$). Then,

- for $X_k \in \mathbb{C}$, $\psi_v^t$ specifies the parameters of a multinomial distribution. The dimension of $\psi_v^t$ is the number of classes in $X_k$;
- for $X_k \in \mathbb{M}$, $\psi_v^t$ specifies the parameters of a two-component gaussian mixture model (GMM) to possibly model either increasing or decreasing $x_k$. Empirically, we find GMM outperforms a single gaussian distribution. The parameter $\psi_v^t$ includes $g_1, g_2, \mu_1, \mu_2, \sigma_1, \sigma_2$. We sample $v^t$ from the density $f_{v^t|w^t}(v^t) = \sum_{k=1}^{2} g_k \frac{1}{\sqrt{2\pi\sigma_k^2}} \exp{-\frac{(v^t - \mu_k)^2}{2\sigma_k^2}}$.

**Baseline function:** The function $b(s^t)$ estimates $\mathbb{E}(R^t(s^t))$, used to stabilize the training process.

**Objective function:** The first objective for $\Pi$ is to make sure the classifier output the target class. We optimize the cumulative reward $R^t$ derived from the theory of REINFORCE (Williams, 1992).

$$\begin{aligned} L_{AR}^t &= -(R^t - b^t) \log p_{\psi_w^t, \psi_v^t}(a^t \mid s^t) \\ &= -(R^t - b^t) \log p_{\psi_w^t}(w^t \mid s^t) - (R^t - b^t) \log p_{\psi_v^t}(v^t \mid w^t, s^t). \end{aligned} \tag{15}$$

Intuitively, when observing state $s^t$, if the action $a^t$ corresponds to a positive cumulative reward compared to baseline $(R^t - b^t)$, we should increase the log probability of choosing $a^t$.

The second objective is to make sure the action pair $(w^t = X_k, v^t)$ forms an conditional intervention $do(X_k = v^t \mid pa_k^t)$, as in definition 2. From Eq. 10 and Eq. 8,

$$L_{cau}^t = \begin{cases} IC(do(X_k \sim \psi_w^t \mid pa_k^t), pa_k^t) & \text{if } X_k \in \mathbb{C}, \\ IC(do(X_k = v^t \mid pa_k^t), pa_k^t), & \text{if } X_k \in \mathbb{M}. \end{cases} \tag{16}$$

Besides, we encourage $\Pi$ to explore diverse actions by adding $L_{entro}^t$, the negative entropy of the policy distribution to the objective function. If $w^t \in \mathbb{M}$, $V^t$ is modeled with a GMM and has no closed-form entropy. Hence, we maximize the lower bound of its entropy (Huber et al., 2008). Lastly, the objective function for $b(s^t)$ is the square error $L_b^t = (R^t - b^t)^2$. The total objective function for the agent is $L_{agent} = \Sigma_{t=0}^{T-1} L_{AR}^t + L_b^t + \beta L_{entro}^t + \eta L_{cau}^t$.

**Alleviating the instability**  Intuitively and empirically, the key of our model to alleviating the instability issue of PASVG(0) is by direct sampling of $w^t$ from $\psi_w^t$ and passing $\psi_w^t$ to $\text{MLP}_v$ to retain the gradient, instead of using Gumbel-Softmax (Jang et al., 2017). Note that to correctly model $P(V^t \mid W^t = w^t)$, $\text{MLP}_v$ requires a clear representation of $w^t$. Sampling using Gumbel-Softmax leads to an obscure representation of $w^t$. Although this can be solved by straight-through trick (Jang et al., 2017), it induces another problem – biased gradient, making training very difficult.

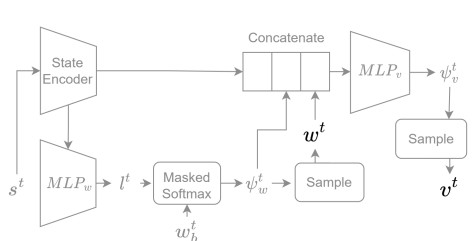

Figure 4: Model architecture of the RL agent.

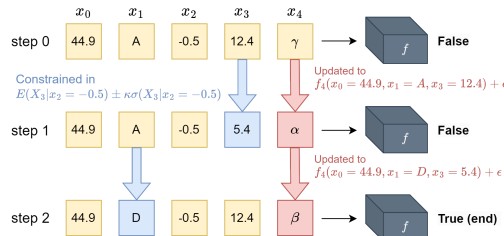

Figure 5: An example of CIAR. The causal graph is based on Figure 2. Blue arrows: interventions by the agent. Red arrows: the treatment effects caused by the interventions.

**Example:**  Figure 5 is a real case example based on the causal model in Figure 2. At step 0, $w^0 = X_3$, the agent takes an conditional intervention $do(X_3 = 5.4 \mid x_2^0 = -0.5)$ since in figure 2, feature $X_3$ has a parent feature $X_2$. The value $v^t = 5.4$ is constrained within $\mu(X_3 \mid x_2 = -0.5) \pm \kappa\sigma(X_3 \mid x_2 = -0.5)$ due to the intervention cost. On the other hand, $X_4$ is a descendant of $X_3$, the RL transition function enforces the treatment effect of $do(X_3 = 5.4 \mid x_2^0 = -0.5)$ to $X_4$. The process continues until the classifier outputs the target class or all features are intervened upon.

## 4 EVALUATION

We compare CIAR to other causality-based AR/ CE methods on three datasets and six metrics that reflect feasibility of ARs in the real world. The formulae of the metrics, the details of datasets and the training environments are left to the Appendix.

### 4.1 METRICS

**Validity:** The score is the proportion of counterfactual instances that truly change the classifier output $f(\cdot)$ to achieve the desired outcome.
**Causal edge score (CES):** CES reflects the preservation of causality between the endogenous features and their parents. CES is originally proposed by Mahajan et al. (2019) while we slightly modify the formula to adjust the score range. CES represents the likelihood ratio of between the counterfactual and the original instances. Since CES can be affected by extreme values, we also report median.
**Constraint:** The score measures the proportion of counterfactual instances that satisfy pre-defined constraints. Each counterfactual instance is counted valid only when all the constraints are satisfied.
**Proximity:** It measures the L1 distance between original and intervened numerical features.

Table 1: Performance of each method in the metrics. (A) denotes considering interventions only.

| Data | #Test | Method | Validity | CES (median) | Constraint | Prox. (A) | Spars-num (A) | Spars-cat (A) | Time |
|------|-------|--------|----------|--------------|------------|-----------|---------------|---------------|------|
| Syn | 405 | DG-SCM | 0.995 | -0.195 (-0.214) | NA | **0.44 (0.45)** | 0 (0) | 0.36 (0.53) | 87 |
| | | OrdCE | **1.0** | -0.583 (-0.371) | | 0.57 (0.48) | 0.26 (0.26) | **0.99** (0.69) | 3354 |
| | | CIAR | 0.998 | **-0.186 (0.012)** | | 0.47 (0.46) | **0.65 (0.65)** | 0.64 (**0.90**) | **7** |
| San | 666 | DG-SCM | 0.991 | 0.204 (0.086) | NA | **0.64** (0.47) | 0 (0) | NA | 331 |
| | | OrdCE | 0.719 | -1.614 (-1.396) | | 1.21 (1.46) | 0.48 (0.04) | | 11998 |
| | | CIAR | **1.0** | **0.300 (0.200)** | | 1.11 (0.27) | 0.13 (**0.80**) | | **13** |
| Adu | 3628 | DG-SCM | 0.690 | | 0.69 | 1.09 (1.09) | 0.51 (0.49) | 0.73 (0.73) | 11822 |
| | | OrdCE | 0.991 | NA | 0.49 | 1.76 (1.78) | 0.30 (0.24) | **0.89 (0.89)** | 147714 |
| | | FastAR | 0.259 | | **1.0** | **0.01 (0.01)** | **0.99 (0.99)** | 0.86 (0.86) | 2308 |
| | | CIAR | **0.994** | | **1.0** | 1.87 (1.77) | 0.37 (0.59) | 0.85 (0.85) | **164** |

**Sparsity:** It measures the proportion of features in the counterfactual instances that remain unchanged. Sparsity is computed for numerical and categorical features respectively.

**Time:** The total inference time in testing.

Given an SCM or causal constraints, we also compute proximity and sparsity of interventions (i.e. the treatment effects are considered *natural*, so not counted), denoted by (A). Proximity (A) can be higher than regular proximity if an AR method does not reflect the treatment effects of interventions.

## 4.2 DATASETS, BASELINES AND EXPERIMENT SETUP

We include two real world datasets **Sangiovese (San)** (Magrini et al., 2017) and **Adult (Adu)** (Dua & Graff, 2017), for evaluation. Different from the original setting, we drop a feature *CapitalGain* in Adu since we find generally any method can simply increase it to change the classifier output, making the solutions too trivial. We also generate a synthetic **(Syn)** dataset according to the causal model in Figure 2. The generation process is provided in the Appendix.

We compare CIAR with methods that utilize SCMs. We include FastAR (Verma et al., 2022) and OrdCE (Kanamori et al., 2021) introduced in Section 2.2 as competitors. An issue of FastAR is that in a graph allowing multi-hop propagation of treament effect, the node order would be unclear. Therefore, we only evaluate FastAR on Adult.

For more comprehensive evaluation, we propose a variant of DiCE-Genetic (DG) (Mothilal et al., 2020) which is genetic algorithm method. Specifically, we combine DG with the causal proximity from Mahajan et al. (2019) to introduce a SCM. We call the variant DiCE-Genetic-SCM (DG-SCM).

We evaluate all the AR/ CE methods with a Multi-layer perception classifier (Pedregosa et al., 2011) which has one hidden-layer and is trained with Adam (Kingma & Ba, 2017) optimizer. Regarding the synthetic dataset, since OrdCE does not support non-linear SCM, we obtain an approximate linear SCM via a solver (Shimizu et al., 2011) built in OrdCE.

## 4.3 RESULTS

Table 1 shows the performance of each algorithm. We summarize our observations as follows.

**Validity:** CIAR reaches high validity across all datasets. We attribute this to the design of the architecture. Specifically, CIAR relates the decision of *which feature* and *what value* in one policy network. It is designed for both numerical and categorical features without unifying the action space. It may be worth noting that the performance of FastAR is inconsistent with the original paper, which is due to our more challenging settings introduced in Section 4.2.

**Causality and constraints:** CIAR outperforms other baselines in CES and constraint. In Sangiovese, the difference between CIAR and DG-SCM is relatively small because Sangiovese, as an conditional linear gaussian network, exactly meets the required assumptions of the causal proximity from Mahajan et al. (2019). Nonetheless, in the synthetic dataset, DG-SCM performs poorly in CES (median). OrdCE does not perform well in CES and constraints due to the fact that some descendants of a intervened feature may by fixed, not reflecting the treatment effects. FastAR is perfect in satisfying constraints in Adult. Note that the constraint score is only computed for the counterfactual instances that do change the classifier output, i.e., 25.9% of all instances for FastAR.

**Proximity and sparsity:** Considering validity, CIAR provides acceptable proximity and high sparsity, especially when we discuss the intervention (but treatment effects) only. The high sparsity is due to

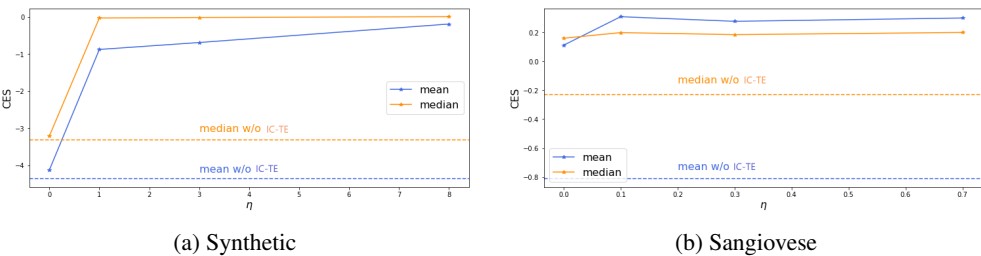

(a) Synthetic        (b) Sangiovese

Figure 6: Causal-edge socre (CES) at different settings of IC and TE.

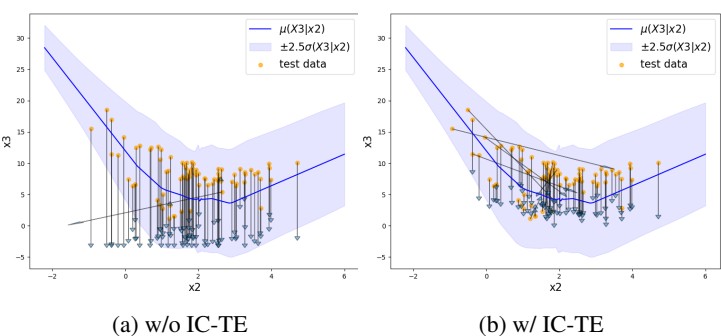

(a) w/o IC-TE        (b) w/ IC-TE

Figure 7: The intervention patterns on $X_2, X_3$ in the synthetic dataset w/ or w/o IC-TE.

the fact that CIAR intervenes upon one feature at each step.

**Speed:** FastAR and CIAR are based on RL while CIAR is much faster. The difference is that CIAR only needs one step to intervene upon a feature. On the other hand, FastAR takes several steps to modify a feature because the continuous action space is discretized.

## 5 DISCUSSION

**Ablation Study**    Figure 6 shows the effect of intervention cost (IC) and treatment effects (TE) on causal-edge score (CES). $\eta$ is the scale of IC; the dashed line represents the CES when neither IC nor TE works. The jump from the dashed line to point point at $\eta = 0$ indicates the pure effect of TE. With $\eta$ going larger, IC kicks in and CES goes higher, showing the effectiveness of IC on CES. IC and TE together form an effective mechanism for preserving feature causality.

**Intervention patterns with IC-TE**    Figure 7 visualizes the effect of IC-TE in the synthetic dataset from 2. In this case, $X_3$ is an important predictor for the classifier. Smaller values of $X_3$ lead to higher probability of target classifier output. Without IC-TE, the RL agent pushes most of the test data to the minimum values in $x_3 - axis$. With IC-TE, since $X_2$ is the parent of $X_3$, when we intervene upon $X_2$, we also update $X_3$ by Eq. 11. When we intervene upon $X_3$, we are actual performing an causal intervention $do(X_3 = v^t \mid x_2^t)$, which confines $v^t$ within a suitable range $\mu(X_3 \mid x_2) \pm 2.5\sigma(X_3 \mid x_2)$(the light blue area). It can be seen that for $x_2 < 0$, $\sigma(X_3 \mid x_2)$ becomes smaller, so there's not much space we can intervene upon $X_3$ directly. Therefore, the RL agent seek to increase the value of $X_2$, which then has a treatment effect (decrease) on $X_3$. It can further intervene upon $X_3$ if needed, but IC will still constrain $X_3$ within a suitable range given $x_2$.

## 6 CONCLUSION

In this work, we propose CIAR, a novel reinforcement learning-based algorithm able to find practical ARs efficiently. We demonstrate that CIAR is especially superior to existing methods in preserving causality. The effectiveness mainly comes from the theoretically preferable loss function and the stable policy network. Furthermore, the policy network is designed to have a bounded number of querying the classifier, which greatly improves efficiency at the inference stage.

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

## A  ESTIMATOR OF CONDITIONAL VARIANCE

From the equation

$$\text{Var}(X_i \mid pa_i) = \mathbb{E}(X_i^2 \mid pa_i) - \mathbb{E}^2(X_i \mid pa_i), \tag{17}$$

we can estimate $\text{Var}(X_i \mid pa_i)$ from the second moment $\mathbb{E}(X_i^2 \mid pa_i)$ and square of the first moment $\mathbb{E}^2(X_i \mid pa_i)$.

Assuming we already have $\mathbb{E}(X_i \mid pa_i)$, then we should estimate $\mathbb{E}(X_i^2 \mid pa_i)$. However, we need to enforce a constraint

$$\text{Var}(X_i \mid pa_i) \geq 0. \tag{18}$$

Instead of predicting $\mathbb{E}(X_i^2 \mid pa_i)$ with this constraint, it is easier to predict $\text{Var}(X_i \mid pa_i)$ with a neural network $\text{MLP}_{var}(pa_i)$ with non-negative activation and minimize the loss

$$(\text{MLP}_{var}(pa_i) + \mathbb{E}^2(X_i \mid pa_i) - x_i^2)^2. \tag{19}$$

## B  EXPERIMENTAL DETAILS

### B.1  METRICS

Since if an counterfactual instance does not actually change the classifier output to the target, then CES, Proximity, sparsity are meaningless for them. Therefore CES, Proximity, sparsity are only computed for CE that really change the classifier output to the target. Let $x_{\cdot,n}$, $x_{num,n}$, $x_{cat,n}$, $x_{v,n}$ denotes the $n^{th}$ (row in dataset) feature vector, numerical feature vector, categorical feature vector and $v^{th}$ dimension feature respectively. $H$ is the set of all endogenous features. $N$ is the size of a dataset and $p$ is the number of features. $\mathbb{1}(x_{\cdot,n}) = 1$ if $f(x_{\cdot,n}) = y'$ else 0.

**Validity**  Validity is the proportion of CE that truly change the classifier output $f(\cdot)$ to the target $y'$.

$$Validity = \frac{1}{N} \sum_{n=1}^{N} \mathbb{1}(x'_{\cdot,n}). \tag{20}$$

**Causal edge score (CES)**  CES reflects the preservation of causality between the endogenous features and their parents. We design a new version of CES from Mahajan et al. (2019). This version represents a log likelihood ratio, which is more common in statistics. Since CES can be affected by extreme values, we also report the median of CES.

$$\frac{1}{|H| \sum_{n=1}^{N} \mathbb{1}(x_{\cdot,n})} \sum_{n=1}^{N} \sum_{v \in H} \log \frac{p\left(x'_{v,n} \mid pa(x_{v,n})'\right)}{p\left(x_{v,n} \mid pa(x_{v,n})\right)} \mathbb{1}(x_{\cdot,n}). \tag{21}$$

For conditional distribution that cannot be expressed by closed-form function, we estimate it with kernel density estimation in Statsmodels Seabold & Perktold (2010).

**Constraint**   Constraint measures the proportion of CEs that satisfy the pre-defined constraints. In our experiments, there are three kinds of constraints. Each CE instance is counted valid only when all constraints are satisfied.

- Feature A is not actionable.

- Feature A can cannot increase (decrease).

- Feature A increases $\implies$ Feature B increases.

**Proximity**   Proximity measures the difference between the original numerical features and perturbed numerical feature. Proximity is calculated under standardized scale.

$$\frac{1}{\sum_{n=1}^{N} \mathbb{1}(x_{\cdot,n})} \sum_{n=1}^{N} \|x_{num,n} - x'_{num,n}\|_1 \mathbb{1}(x_{\cdot,n}). \tag{22}$$

**Sparsity**   Sparsity measures the proportion of features in CE that remain unchanged. Sparsity is computed for numerical and categorical features respectively.

$$1 - p\frac{1}{\sum_{n=1}^{N} \mathbb{1}(x_{num/cat,n})} \sum_{n=1}^{N} \|x_{\cdot,n} - x'_{\cdot,n}\|_0 \mathbb{1}(x_{\cdot,n}). \tag{23}$$

## B.2  DATASETS

**Synthetic**   We generate a dataset with 8 features and a binary outcome according to the causal model in Figure 2. $X_1, X_4$ and $U_2$ are categorical and the rest are numerical. All features are assumed actionable. The data generation process is shown in the followings.

$$\begin{aligned}
X_0 :=& \mathcal{N}(50, 15) \\
X_1 :=& Multinomial(0.3, 0.2, 0.25, 0.25) \\
X_2 :=& \mathcal{N}(2, 1) \\
U_0 :=& \sqrt{|X_2|} + \mathcal{N}(2, 1) \\
U_1 :=& \mathcal{N}(0, 0.3) \\
U_2 :=& Bernoulli(0.5) \\
X_3 :=& (X_2 - 3)^2 + \mathcal{N}(-3, 0.5) * U_2 + \mathcal{N}(3.5, 0.5) * (1 - U_2) + U_0 \\
X_4 :=& Multinomial(p_1, p_2, p_3)
\end{aligned} \tag{24}$$

where

$$\begin{aligned}
p_1 :=& |X_0|/100 + 0.1 * \mathbb{1}(X_1 = 0) + 0.1 * \mathbb{1}(X_1 = 2) * \sqrt{|X_3|} \\
p_2 :=& 0.3 * \mathbb{1}(X_1 = 1) + 0.4 * \mathbb{1}(X_1 = 4) + X_3/10 \\
p_3 :=& X_0 * X_3/1000 + 0.1
\end{aligned} \tag{25}$$

We drop three features $U_0, U_1$ and $U_2$ to simulate the situation of unobserved parents of the endogenous features.

**Sangiovese (Magrini et al., 2017)**   Sangiovese is a dataset with 13 numerical features and one categorical features. (Magrini et al., 2017) fit a conditional linear gaussian network Lauritzen & Spiegelhalter (1988) on the dataset with some expert knowledge. Like in (Mahajan et al., 2019), we drop the categorical feature for simplicity. All variables are assumed actionable. The pre-processed dataset is downloaded from `https://github.com/divyat09/cf-feasibility`.

**Adult (Dua & Graff, 2017)**   Adult contains numerical and categorical features. We drop a feature *CapitalGain* since if it exists, almost every method choose to simply increase it to change the classifier output. The solution then becomes trivial. Dropping *CapitalGain* makes the problem much more challenging. The pre-processed dataset is downloaded from `https://github.com/vsahil/FastAR-RL-for-generating-AR`.

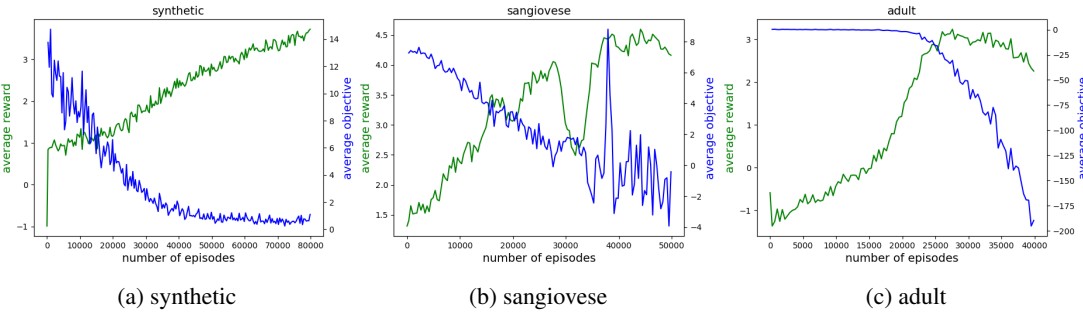

Figure 8: The learning curve of each dataset.

## B.3 EXPERIMENTAL RESOURCES

All experiments run on Intel Xeon cpu E5-2650v4@2.20GHz. In addition to the inference time reported in Table 1, CIAR and FastAR Verma et al. (2022) require training. It takes about 50, 20 and 40 minutes to train CIAR on the synthetic, sangiovese and adult datasets respectively. It takes about 2 hours to train FastAR on Adult dataset.

## B.4 ALGORITHM-SPECIFIC SETTING

**CIAR**  The discount factor $\gamma$ is set to 0.95. $\lambda$ (that controls regularization), $\beta$ (that controls exploration) and $\eta$ (that controls the scale of intervention cost) are treated as hyperparameters and tuned manually.

**OrdCE (Kanamori et al., 2021)**  The interaction matrices for Sangiovese and Adult dataset are given. Since it does not support non-linear causal model, we solved a linear causal model using OrdCE's built-in causal discovery method (Shimizu et al., 2011). The optimization process takes too long, so we set the time limit to 10 seconds for each instance, but the solver may not really respond to the setting in some cases.

**Fastar (Verma et al., 2022)**  Categorical features, like in the original implementation, are treated as numerical. We round the value to its closest integer-encoded category and then one-hot encode it, so it can be fed to the classifier.

## B.5 LEARNING CURVES

Figure 8 shows the learning curves of each dataset. Each point represents the average result of 64 random samples.

## C ADDITIONAL EXPERIMENTS

### C.1 STUDY OF REPLACING GUMBEL-SOFTMAX

Table 2 shows the performance of CIAR with direct sampling or with gumbel-softmax. The results are close. However, during our experiments, we observed that using gumbel-softmax could occasionally cause model weights to go to Not-a-Number error when running on Sangiovese dataset.

Table 2: Performance of CIAR with direct sampling or with gumbel-softmax. CIAR-G denotes the gumbel-softmax version.

| Data | #Test | Method | Validity | CES (median) | Constraint | Prox. (A) | Spars-num (A) | Spars-cat (A) | Time |
|------|-------|--------|----------|--------------|------------|-----------|---------------|---------------|------|
| Syn | 405 | CIAR | 0.998 | -0.186 (0.012) | NA | 0.47 (0.46) | 0.65 (0.65) | 0.64 (0.90) | 7 |
|     |     | CIAR-G | 0.993 | -0.283 (0.093) |    | 0.51 (0.50) | 0.64 (0.65) | 0.62 (0.87) | 7 |
| San | 666 | CIAR | 1.0 | 0.300 (0.200) | NA | 1.11 (0.27) | 0.13 (0.80) | NA | 13 |
|     |     | CIAR-G | 1.0 | 0.489 (0.267) |    | 1.24 (0.20) | 0.01 (0.91) |    | 9 |
| Adu | 3628 | CIAR | 0.994 | NA | 1.0 | 1.87 (1.77) | 0.37 (0.59) | 0.85 (0.85) | 164 |
|     |      | CIAR-G | 0.996 |    | 1.0 | 2.82 (2.79) | 0.48 (0.54) | 0.87 (0.87) | 163 |

