# OpenReview forum: "Fast Conditional Intervention in Algorithmic Recourse with Reinforcement Learning"
_ICLR.cc/2024/Conference — Submitted to ICLR 2024_

### Official Review · Reviewer_a8gS · 2023-10-28

**Soundness:** 2 fair
**Presentation:** 3 good
**Contribution:** 2 fair
**Rating:** 5
**Confidence:** 3

**Summary:**

The paper proposes an RL-based method for the recourse generation problem. The paper incorporates causal graphs of input features to calculate a new cost for conditional intervention called Intervention Cost. The experiments conducted on synthetic and real-world datasets show a better performance than baselines.

**Strengths:**

- The paper is easy to read and follow.
- The construction of the Intervention Cost is sound and highly motivated.
- The proposed Markov Decision Process is well-defined and reasonable.

**Weaknesses:**

- The paper assumes that all subjects share the same prior causal graph. However, in reality, each individual typically possesses a distinct causal graph. To address this concern, De Toni et al. (2022) [2] propose a solution. They initially establish a fixed causal graph and then iteratively learn the subject-specific cost function. Subsequently, they seek an appropriate sequence of interventions.
- The author omits the description of the reinforcement learning algorithm used to solve the MDP and its parameters.
- The way the author handles the noisy graphs (incompleteness of the casual graph) is unclear.
- The learning curve of rewards, objectives, and metrics should be reported. The evaluation can be improved by comparing the proposed method and baselines on more datasets.
- In Section 3.2.3, the authors state that architectural corrections can alleviate the instability of the PASVG(0). However, there is no justification or ablation study for this claim.

**Questions:**

- In section 3.2.2, when finding the longest path length between $X_i$ and $X_k$, what is the edge weight between two vertices of the graph?  Does the algorithm find the longest path on the casual graph?
- The reward function and the objective function in Section 3.2.2 are not related to each other, making me confused about interpreting their role in the training.

**References**

[1] Sahil Verma, Varich Boonsanong, Minh Hoang, Keegan E. Hines, John P. Dickerson, and Chirag Shah. Counterfactual explanations and algorithmic recourses for machine learning: A review, 2020.

[2] Giovanni De Toni, Paolo Viappiani, Bruno Lepri, and Andrea Passerini. Generating personalized counterfactual interventions for algorithmic recourse by eliciting user preferences, 2022.

---

> ### Author Response · Authors · 2023-11-21
>
> We appreciate your precious comments. Hopefully the followings will address your concerns.\
> \
> **Response to weakness 1** \
> We agree that the idea from De Toni et al. is interesting. However, we would also like to highlight the difference and connection between these two works. Different from De Toni et al. who propose to incorporate user preferences, we focus on improving recourse by considering causality and thus do not require human-in-the-loop. It is true that our model does not personalize recourse; however, we find no apparent conflict of introducing the cost function proposed by De Toni et al. to our model. Specially, given that the two works both consider uncertainty between features, it may be possible to find a probabilistic approach combining the merits (i.e., conditional intervention and personalization) concerning distinct issues.\
> \
> **Response to weakness 2** \
> The specification of MDP is in Section 3.2.2, and the objective function of the reinforcement learning algorithm, REINFORCE (Williams, 1992), is shown in Eq. 15. We also revise the paper to make the description more clear. The newly added paragraph “Intuitively, when observing state $s^t$ (current feature values), if the action $a^t$ corresponds to a positive cumulative reward compared to baseline $(R^t-b^t)$, we should increase the log probability of choosing $a^t$” is in Section 3.2.3. Finally, we would like to emphasize that the objective function is only for the classifier to output the target class. The total cost we optimize is $L_{agent}$, in which the parameters account for the degree of exploration ($\beta$) and the strength of intervention cost ($\eta$).\
> \
> **Response to weakness 3** \
> In our work, incompleteness refers to the unobserved variables and we propose to model the incompleteness via variance as discussed in Section 1. Let us say we have a simple causal graph A → B where there is no unobserved variable. In this case, the estimated variance of $P(B | A)$ shall be near 0 and our model will then choose to intervene A when we need a different value of B. On the contrary, if there exists an unobserved variable C being a parent of B, our model would discover a higher variance in $P(B|A)$ and therefore conclude the existence of C. Our model would then propose to intervene B directly, revealing the fact that changing A might not be an effective means to affect B. \
> \
> **Response to weakness 4** \
> We now report the learning curves in Appendix B.2. \
> \
> **Response to weakness 5** \
> We now report the performance with and without our trick in Appendix C. The performance gaps are small in general; however, we found using gumbel-softmax can occasionally cause not-a-number errors on Sangiovese dataset and thus require more careful tuning. \
> \
> **Answer to Q1** \
> Yes, the algorithm finds the longest path (LP) on the causal graph. The weight of each edge is set to 1. Consider a causal graph with two paths, A->B->C and A->E->F->C. Supposed at one step, the RL agent chose to intervene upon feature A, then firstly B, E (LP(A,B)=LP(A,E)=1) should update to new values because of the treatment effect from A. Secondly, F should update (LP(A,F)=2) because of the treatment effect of E.  Finally C should update (LP(A,F)=3) because of the treatment effects from B and F.   \
> \
> **Answer to Q2** \
> We are sorry for the confusion.The reward we adopt is the cumulative reward $R^t$ rather than reward $r^t$, which appears following eq. 12. This is because the action at step t will have effects on all the steps after t. Therefore, in the objective function (eq. 15), the probability of choosing action $a^t$ is updated according to the cumulative reward $R^t$.

---

> > ### Comment · Reviewer_a8gS · 2023-11-22
> >
> > I appreciate the author's response. The answer has clarified lots of aspects of the paper. I have raised the score from 3 to 5.

---

### Official Review · Reviewer_J2Mk · 2023-10-30

**Soundness:** 3 good
**Presentation:** 4 excellent
**Contribution:** 2 fair
**Rating:** 5
**Confidence:** 3

**Summary:**

The paper proposes to use RL agent for helping design more efficient and accuracy intervention strategies for explanations.  By the desgined architecture with the so-called interventional cost as loss functions, the method shows some advantage over existing ones on some datasets.

**Strengths:**

1. The use of RL is interesting.
2. The experiments concerning interventions are convincing.

**Weaknesses:**

1. Some theoretical properties need more justifications.
2. The efficiency of training needs more evaluations.

**Questions:**

1. About Fig 2. Is this graph representative? It seems the only confounder is U_0, and other Us can be considered as additive noise. Why this graph is used as an example for experiments?
2. About the theoretical aspects of "incomplete SCM". Is there any theoretical justification of how "incomplete" your method works? Or under some quantification of missing nodes, can you show some error bounds or something like that?
3. About the RL part. Is there anything related to the choice of reward, policy that have impacts on the final experimental outcomes?

---

> ### Author Response · Authors · 2023-11-21
>
> We appreciate your precious comments. Hopefully the followings will address your concerns.\
> \
> **Response to weakness** \
>  Our model and FastAR require training, and the training time of the 2 models are 10 - 50 minutes and 2 hours on Intel Xeon cpu E5-2650-v4. We leave the details to Appendix B.3. \
> \
> **Answer to Q1** \
>  In this work, we assume the existence of unobserved variables in the causal graph and thus design Figure 2 as an example for illustrating the idea. Also, we generate synthetic data via nonlinear equations (Appendix B.2) for the experiments in Section 4. Notably, we assume the unobserved variables do not cause spurious correlations. Therefore, we design $U_0$ pointing to one endogenous feature and $U_1$ on the path of observed features as the examples. \
> \
> **Answer to Q2** \
> We quantify the incompleteness or uncertainty via estimated variances (numerical) and entropy (categorical). We assume that higher variance or entropy is a stronger signal of missing causes of an endogenous feature. The proposed Intervention cost (IC) is a generalization of the causal proximity $\sum_v \|f_v(pa_v’) - x_v’\| \ v\in endo$ from Mahanjan et al. [1]. When all the endogenous features $X_{vs}$ share the same constant conditional variance (given parent feature values $pa_v’$), IC degrades to the causal proximity from Mahajan et al. IC should bound the endogenous features $X_{vs}$ in the region given by eq. 7. The error (deviation from eq. 7) depends on the difficulty of changing the given classifier output, which is our primary goal. Usually the more difficult it is, there can be more deviation from eq. 7 for the endogenous features since it requires more and larger interventions on them. \
> \
> **Answer to Q3** \
> There are two terms in the reward (eq. 12), one for changing the classifier output; the other for controlling the proximity. The parameter $\lambda$ in eq.12 decides the degree of controlling the proximity. There is usually a tradeoff between validity and proximity since some difficult cases require larger interventions to change the classifier output.\
> \
> As for the policy, we find it better to model the distribution of a numerical feature with a two-component GMM than with a simple gaussian distribution. For example, for a simple logistic regression model $f(x)=\frac{1}{1+e^{-x^2}}$, when $x=0$, increasing or decreasing $x$ both increases the probability of a positive output. A learnable GMM is able to preserve the possibility of the two actions and degrade to a single gaussian distribution when facing simple cases.
>  \
> \
> [1] Divyat Mahajan, Chenhao Tan, and Amit Sharma. Preserving causal constraints in counterfactual explanations for machine learning classifiers. 2019.

---

> > ### Comment · Reviewer_J2Mk · 2023-11-22
> > **Thank you for the comments**
> >
> > Thank you for the Rebuttal. I still think the paper contains several novel ideas but the limitation from theoretical aspect is also present, and remain unchanged of my score.

---

### Official Review · Reviewer_vwWM · 2023-11-02

**Soundness:** 2 fair
**Presentation:** 2 fair
**Contribution:** 3 good
**Rating:** 6
**Confidence:** 4

**Summary:**

This paper addresses the problem of finding realistic and causally-grounded counterfactual explanations. They propose a reinforcement learning (RL)-based approach with conditional interventions. The proposed intervention method has theoretical properties, e.g., it considers both feature dependencies leveraging the SCM. For the RL strategy, computational complexity is provided. Experiments are performed on synthetic and real datasets.

**Strengths:**

This paper brings together counterfactual fairness, causality, and reinforcement learning.
The strategy tries out several interventions using reinforcement learning to identify a realistic recourse given an SCM. It is mathematically interesting.

The challenge arises since at each stage the RL agent has to decide which feature to intervene and also with what value. To address this challenge, the RL agent will leverage a structural causal model. Then, it would perform conditional interventions, i.e., interventions conditioned on the parents of that feature. Ultimately, the goal is to obtain a counterfactual that will respect the SCM and also be as close to the original point as possible in fewer steps than the number of features changed. Additionally, they require the number of interventions T to be less than p which is the number of actionable features.

They have included relevant baselines in their experiments, and show time benefits.

**Weaknesses:**

One limitation is that the SCM may not always be available.

The scenario of incomplete causal graphs as mentioned in the abstract was not very clear to me. What is the assumption here?

The experiments directly seem to use the causal discovery method of another paper. Is this done for the proposed method as well?

I also wonder if RL is a bit of an overkill for this problem since the number of features (p) is often quite small. It is often desirable to intervene on fewer features. For instance, the experiments drop the feature Capital Gain since intervening only on that one feature suffices for recourse. Also, what about exploration? Could the authors strengthen the motivation behind this approach?

And also, how is the time being calculated in the experiments? It seems to be only the inference time. What about preprocessing time? Could the authors discuss/elaborate on the preprocessing time of various methods?

The experiment section does not provide enough details on how the causal graph was generated for the real-world datasets and if that causal graph is reliable.

Ultimately, human evaluations might also be necessary at some point to compare different methods.

**Questions:**

Already discussed in weakness.

---

> ### Author Response · Authors · 2023-11-21
>
> We appreciate your precious comments. Hopefully the followings will address your concerns.\
> \
> **One limitation is that the SCM may not always be available.**\
> We cannot agree more. This is the reason why this work chooses to relax the requirement of needing a complete SCMs. We assume that a partially observed SCM can be obtained much easier in the real-world situation. After all, as discussed by Karimi et al. [1], recourse without SCMs cannot be guaranteed unless more assumptions are introduced.\
> \
> **The scenario of incomplete causal graphs as mentioned in the abstract was not very clear to me. What is the assumption here?** \
> The assumed incompleteness is the unobserved variable set. As shown in Figure 1, we assume an endogenous variable can be influenced by unobserved variables in the graph. A partially observed casual graph is more realistic and easier to obtain.\
> \
> **The experiments directly seem to use the causal discovery method of another paper. Is this done for the proposed method as well?**\
> We do not use causal discovery methods in all experiments except for OrdCE on the synthetic dataset. Since OrdCE is specifically designed for linear SCMs, we employ its built-in causal discovery method to obtain an approximated SCM. For the Sangiovese and Adult datasets where SCMs are linear, we let OrdCE directly use the SCMs. \
> \
> **I also wonder if RL is a bit of an overkill for this problem since the number of features (p) is often quite small. It is often desirable to intervene on fewer features. For instance, the experiments drop the feature Capital Gain since intervening only on that one feature suffices for recourse. Also, what about exploration? Could the authors strengthen the motivation behind this approach?** \
> We believe one of the challenges of suggesting ideal interventions is considering the causations between features. Take figure 1 for example, considering the consequence of changing a variable (i.e., height) helps us improve recourse (i.e., +2 instead of +8 kg). We argue that handling such dependency can be challenging when more features are involved and we thus need an approach flexible and powerful enough to obtain ideal outcomes. \
> \
> Another advantage of our RL approach lies in the efficiency. Optimization-based methods (e.g. DiCE [2]) solve an optimization problem for every instance. The proposed RL-based method, on the other hand,  performs fast inference for new instances after training.\
> \
> **And also, how is the time being calculated in the experiments? It seems to be only the inference time. What about preprocessing time? Could the authors discuss/elaborate on the preprocessing time of various methods?**\
> We finish preprocessing before inference. Since we apply the same preprocessing (normalization) to data for all methods, we did not include preprocessing time for comparison. Also, as reported in Appendix B.3,  our model and FastAR require training, and the training time of the 2 models are 10 - 50 minutes and 2 hours on Intel Xeon cpu E5-2650-v4.\
> \
> **The experiment section does not provide enough details on how the causal graph was generated for the real-world datasets and if that causal graph is reliable.** \
> We are sorry for the missing details. The sources of the causal graphs are as follows.\
> Sangiovese: https://www.bnlearn.com/bnrepository/clgaussian-small.html#sangiovese \
> Adult: We use the pre-processed dataset released by the authors of FastAR [3] (https://github.com/vsahil/FastAR-RL-for-generating-AR). \
> Synthetic: We report the structure and equations in Appendix B.2 for examination.\
> \
> **Ultimately, human evaluations might also be necessary at some point to compare different methods.**\
> Indeed. We thank and agree with the reviewer for this suggestion.
>
>  [1] Amir-Hossein Karimi, Bodo Julius von Kügelgen, Bernhard Schölkopf, and Isabel Valera. Algorithmic recourse under imperfect causal knowledge: a probabilistic approach. 2020.\
> [2] Ramaravind K. Mothilal, Amit Sharma, and Chenhao Tan. Explaining machine learning classifiers through diverse counterfactual explanations. 2020.\
> [3] Sahil Verma, Keegan Hines, and John P. Dickerson. Amortized generation of sequential algorithmic recourses for black-box models. 2022.

---

### Official Review · Reviewer_1q46 · 2023-11-08

**Soundness:** 3 good
**Presentation:** 2 fair
**Contribution:** 3 good
**Rating:** 5
**Confidence:** 3

**Summary:**

The paper proposes an efficient RL-based approach with the idea of conditional intervention, with the goal of handling noisy and/or incomplete graphs, as well as efficient performance of inference for black-box classifier. The experimental results show the efficiency of the proposed method on both synthetic and real datasets.

**Strengths:**

The paper tackles an important problem in algorithmic recourse, which is causal sequential recourse, using the technique from reinforcement learning that works in a boarder setting compared to the previous paper.

**Weaknesses:**

One weakness of the paper is the assumptions are pretty strong-- it feels like a lot of assumptions (e.g., the formulation of intervention cost) are made for mathematical convenience rather than for accurate modeling. In addition, the writing and structure of the paper can be improved; for example, it is still unclear to me how CIR is especially superior to existing methods in preserving causality and how the method handles incomplete graph cases. Answering the questions in the Questions section might help make some clarifications.

**Questions:**

1. The paper mentions that "The less it is determined by their parents, the more _space_ we can intervene." Could you explain more why that's the case? in particular, what does "space" mean? And why do we want to primarily intervene in higher uncertainty endogenous features?

2. Does the size of the action space grow exponentially as a function of the feature space? If so, how does the algorithm handle this?

3. Intuitively, what is the benefit of conditional intervention compared to traditional intervention?


Typo:

1. At the bottom of page 5, "...$X_k$ is intervened upon is calculated by.."

---

> ### Author Response · Authors · 2023-11-21
>
> We appreciate your precious comments.\
> \
> **Response to weakness** \
> Since we optimize the agent with reinforcement learning, it is in fact rather easy to optimize a complex conditional density for an endogenous feature in the design of the intervention cost. However, estimating accurate density conditioning on multiple parents is difficult due to data sparsity. Causal proximity from Mahajan et al. [1] is another extreme, in which only the first moment (conditional mean) is estimated. Our intervention cost (IC) estimates the first and second moment (conditional variance), which is an understandable statistic that describes uncertainty. Also the range constructed by conditional mean and variance is guaranteed to cover the actual distribution to some extent by Chebyshev’s inequaltiy. \
> \
> **Answer to Q1** \
> High uncertainty of an endogenous variable given its parents implies there exists an influential factor that is not observed. Namely, the parents are not sufficient to decide the value of the endogenous variable. Let us assume there are two variables A and B with a relation A → B where A fully decides B. If B needs to be higher to change the classifier output, the ideal suggestion from an AR method would be intervening A instead of B, as it is the only way to control B. On the contrary, if B is highly uncertain given A, there may exist a hidden factor C having relation C → B for intervention. As C is not observed, changing B through C is simplified to changing B in our AR method output. The “space” in this case is the uncertainty of B given A. In this work, we model the uncertainty via variance of $P(B|A)$ introduced in Eq.6 and 8. \
> \
> **Answer to Q2** \
> It is true that the action space of discrete features can grow exponentially. To the best of our knowledge, our competitors all face the same challenge while our model has an advantage of inference speed as shown in the experiments. \
> \
> **Answer to Q3** \
> The main difference is that conditional intervention considers effects from the parents.
> Consider a teenager of 155cm and 70kg to make the basketball team. In the traditional intervention, height and weight are treated independently. It can result in the intervention of +8 kg to make the team, as 70+8kg seems to be a reasonable weight for a person, if the height is not considered. On the other hand, conditional intervention cares about the fact that (70+8) kg is too much for a 155cm person. It could seek for another option “+2cm and + 4kg” while considering both factors together. \
> \
> [1] Divyat Mahajan, Chenhao Tan, and Amit Sharma. Preserving causal constraints in counterfactual explanations for machine learning classifiers. 2019.

---

> > ### Comment · Reviewer_1q46 · 2023-11-21
> > **Thank you for your response**
> >
> > I've carefully read the response from the author, and my evaluation remains the same.

---

### Meta-Review · Area_Chair_MQgQ · 2023-12-12

**Metareview:**

The paper presents a new method for causal recourse provision. The proposed method uses ideas from reinforcement learning to generate feature-level perturbations that can attain a target prediction. Key benefits include speed; handling noisy and/or incomplete graphs; and model agnostic approach. The paper includes experimental results showing the efficiency of the proposed method on both synthetic and real datasets.

**Strengths**

- Topic: the paper presents a new method to generate recourse actions that are robust to causal effects.

**Weaknesses**

- Reliability: The reliability of the method hinges on the assumption that practitioners have access to a causal graph that is correctly specified for all individuals in a target population. This is a relatively strong assumption and should be studied in detail -- the work should examine the potential for misspecification at a population level, and at an individual level (e.g., under heterogeneity that cannot be captured by noise).

- Significance: The work has the potential to make a significant impact - by developing a practical method to address an important problem in recourse provision (e.g., causal after-shocks that invalidate recourse). The manuscript fails to realize this by focusing on issues that appear tangential. Ideally, we would have compelling empirical evidence to highlight the importance of accounting for causality in this setting and the challenges of doing so.

- Soundness: The paper ignores some of the major technical challenges associated with actionability.  The proposed framework is not designed to handle hard actionability constraints on feature-level constraints (e.g., integrality, monotonicity, if-then constraints). In turn, the proposed "interventions" may violate actionability. On the flip side, the method will also fail to flag instances where "interventions" do not exist. The manuscript ignores these challenges.

**What is Missing**

The key issue here is not that the method should be able to handle all these constraints, but that these issues should be studied and addressed given that this paper is about "recourse" (rather than "counterfactual explanations"). At a minimum, readers should know which kinds of actionability constraints the method can support, and how the method works in settings where these constraints cannot be enforced. Currently, these limitations are neither stated nor addressed.  be clear, recourse should be actionable, and recourse  for recourse should be designed to handle the fact that recourse may not exist

**Justification For Why Not Higher Score:**

See review.

**Justification For Why Not Lower Score:**

N/A

---

### Decision · Program_Chairs · 2024-01-16

Reject